# A multisite randomized controlled trial of an early palliative care intervention in children with advanced cancer: The PediQUEST Response Study Protocol

Veronica Dussel [1,2] *, Liliana Orellana [3], Rachel Holder [4], Rachel Porth [5], Madeline Avery [6], Joanne Wolfe [6,7,8,9]

1 Pediatric Palliative Care, Dana-Farber / Boston Children's Cancer and Blood Disorders Center, Boston, MA, United States of America, 2 Center for Research and Implementation in Palliative Care, Institute for Clinical Effectiveness and Health Policy, Buenos Aires, Argentina, 3 Biostatistics Unit, Deakin University, Geelong, Australia, 4 Department of Psychology, Virginia Commonwealth University, Richmond, VA, United States of America, 5 Department of Internal Medicine, Beth Israel Deaconess Medical Center, Boston, MA, United States of America, 6 Department of Psychosocial Oncology and Palliative Care, Dana-Farber Cancer Institute, Boston, MA, United States of America, 7 Department of Pediatric Oncology, Dana-Farber Cancer Institute, Boston, MA, United States of America, 8 Department of Pediatrics, Boston Children's Hospital, Boston, MA, United States of America, 9 Harvard Medical School, Boston, MA, United States of America

* veronica_dussel@dfci.harvard.edu

## Abstract

### Background

The Pediatric Quality of Life and Evaluation of Symptoms Technology Response to Pediatric Oncology Symptom Experience (PQ-Response) intervention aims to integrate specialized pediatric palliative care into the routine care of children, adolescents, and young adults (AYAs) with advanced cancer.

### Aims

To evaluate whether PQ-Response, compared to usual care, improves patient's health related quality of life (HRQoL) and symptom burden (aim 1), parent psychological distress and symptom-related stress (aim 2), and family and symptom treatment activation (aim 3).

### Design

*Multisite, randomized (1:1), controlled, un-blinded, effectiveness trial* comparing Pedi-QUEST Response (intervention) vs usual cancer care (control).

### Setting

Five US large, tertiary level pediatric cancer centers.

### Participants

Children (≥2 years old)/AYAs who receive care at any of the participating sites because of advanced cancer or any progressive/recurrent solid or brain tumor and are palliative care

relevant data from this study will be made available upon study completion.

**Funding:** JW, R01 grant number 5 R01 NR016720-03 funded by the National Institutes of Health National Institute of Nursing Research. https://www.nih.gov/about-nih/what-we-do/nih-almanac/national-institute-nursing-research-ninr The funders had and will not have a role in study design, data collection and analysis, decision to publish, or preparation of the manuscript.

**Competing interests:** The authors have declared that no competing interests exist.

"naïve." Target: 200 enrolled patient-parent dyads (minimum goal: 136 dyads randomized, N = 68/arm).

## Interventions

*PediQUEST Response*: combines patient-mediated activation (weekly feedback of patient- and parent-reported symptoms and HRQoL to families and providers using the PediQUEST web system) with integration of the palliative care team. *Usual Cancer Care*: participants receive usual care, which can include palliative care consultation, and use PediQUEST web to answer surveys, with no feedback.

## Methods

Following enrollment, patients (if ≥5 years) and one parent receive weekly PediQUEST-Surveys assessing HRQoL (Pediatric Quality of Life Inventory 4.0) and symptom burden (PediQUEST-Memorial Symptom Assessment Scale). After a 2-week run-in period, dyads who answer ≥2 PediQUEST surveys per participant (responders), are randomized (concealed allocation) and followed up for 16-weeks. Parents answer six additional surveys (parent outcomes).

## Outcomes

Primary: mean patient HRQoL score over 16-weeks as reported by a) the parent; and b) the patient if ≥5 years-old. Secondary: patient's symptom burden; parent's anxiety, depressive symptoms, symptom-related stress; family activation; and symptom treatment activation.

## Trial registration

ClinicalTrials.gov (NCT03408314) 1/24/18. https://clinicaltrials.gov/ct2/show/NCT03408314.

## Introduction

### Background and rationale

Integration of palliative care (PC) into healthcare through early consultation, education, or symptom monitoring and feedback has been associated with better health related quality of life (HRQoL) and longer survival in adult patients [1–3] and their caregivers [4]. A growing number of randomized controlled trials (RCTs) in this regard led to the American Society for Clinical Oncology's endorsement of early PC integration for adult patients with metastatic cancer and/or high symptom burden [5]. International and national organizations also have called for early PC integration for children [6–8]. However, in a recently published systematic review we did not identify any RCTs that evaluated whether specialized PC improves child and family outcomes [9]. Evidence in favor of pediatric PC mostly comes from non-experimental studies [10–15] and a few RCTs focusing on selected aspects of care [16,17]. Children and teens with advanced cancer endure a high degree of suffering [18,19] linked to impaired patient [20,21] and family [22,23] survivorship experience [11,18,19,24,25]. Yet, PC integration is highly variable in terms of availability [26,27], frequency and timing of PC referrals [28,29] and limited by primary oncologist and family beliefs [30].

In prior work, we identified substantial child suffering from cancer-directed therapies and symptoms [18,31–33]. These findings led to conducting a pilot RCT (the Pediatric Quality of Life and Evaluation of Symptoms Technology (PediQUEST) Study) to evaluate a primary PC intervention consisting of providing symptom and HRQoL data to families and clinicians (collected with an electronic-patient reported outcomes (e-PROMs) system, PediQUEST) [34]. The study showed that: (i) collecting e-PROMs in these children was feasible [35]; (ii) symptom burden was high [19]; (iii) PediQUEST reports increased awareness about distress; and (iv) symptom distress and HRQoL improved although not significantly [34].

As a next step, we developed the PediQUEST Response to the Pediatric Oncology Symptom Experience (PediQUEST/PQ Response) intervention, a strengthened intervention that combines the use of a modernized PediQUEST system and App with early specialized PC intervention for children, adolescents, and young adults (AYAs) with advanced cancer. During the formative research work, we explored barriers and facilitators of the symptom management process and identified that patients, parents, and providers accepted symptoms and suffering as inherent to cancer and its treatment, a pervasive phenomenon that we refer to as "normalization" [36]. This finding led us to focus the PediQUEST Response intervention on activation of both providers and families. We present here the study protocol for a RCT that will assess the effectiveness of the PediQUEST Response intervention. Changes in enrollment and intervention delivery procedures due to the COVID-19 pandemic will be noted when apropriate.

## Study objectives

The overall goal of the study is to conduct a **multisite parallel randomized controlled trial** to evaluate whether the PediQUEST Response intervention improves pediatric oncology patient and parent outcomes compared to usual care.

**Primary objective.** To determine if compared to usual care, PediQUEST Response *improves HRQoL in children* with advanced cancer, as measured through the Pediatric Quality of Life Inventory 4.0 (PedsQL) [37,38] reported weekly over 16-weeks by a) the parent and, b) the patient if ≥5 years of age.

**Secondary objectives.** *Key secondary objectives*. To determine if, when administered to children with advanced cancer, PediQUEST Response is superior to usual care for,

1. improving *child symptom burden*, as measured through the PediQUEST-Memorial Symptom Assessment Scale (PQ-MSAS) [39,40] reported weekly over 16 weeks by a) the parent, and b) the patient if ≥13 years old (age limitation explained in Discussion).

2. improving *parent psychological distress*, as measured by the Spielberger's-State-Trait Anxiety Inventory (S-TAI)-State component [41] and the Center for Epidemiologic Studies Short Depression Scale (CES-D-10) Scale [42], reported every 4 weeks for 16 weeks, and *symptom-related stress*, as measured by an adapted version of the stress portion of the Response to Stress Questionnaires-pain [43] (*a*RSQ-pain) measured at study entry and week 16.

*Other secondary objectives*. We will also assess whether the PediQUEST Response intervention increases *family activation* compared to usual care, by measuring active coping, planning, and instrumental support coping styles using an adapted version of the Brief-COPE scale [44] (at study entry and 16-weeks), and by specifically evaluating symptom treatment activation, measured by the use of integrative strategies for symptom treatment (total number, and number of different, integrative therapies as reported by parents every four weeks, and number of psychosocial clinician encounters documented in the medical record).

## Trial design

This is a randomized, controlled, un-blinded, multisite, superiority effectiveness trial with two parallel groups (1:1), PQ-Response (intervention) vs. usual cancer care (control).

## Methods

### Participants, interventions, and outcomes

**Study setting.**   The study is conducted at five U.S. tertiary level pediatric hospitals that (a) have an established PC interprofessional program with Hospice and Palliative Medicine Board certified physician specialists, (b) treat over 200 newly diagnosed cancer patients per year, and (c) have no early PC integration program in place. Participating sites include Dana-Farber/ Boston Children's Cancer and Blood Disorders Center (DFBCC), coordinating center, Children's Hospital of Philadelphia Cancer Center (CHOP), (PI: Feudtner), Ann and Robert H. Lurie Children's Hospital of Chicago (PI: Waldman), Seattle Children's Hospital (SCH), (PI: Rosenberg), and Texas Children's Hospital (PI: Kang).

**Eligibility criteria.**   We intend to enroll a total of 200 dyads to achieve our minimum target sample of 150 randomized and 136 with complete data. Enrollment started in April 2018. We are recruiting consecutive patients along with one of their parents until the target sample is reached. Table 1 presents inclusion and exclusion criteria. The study protocol was originally approved by the Dana-Farber/Harvard Cancer Center Institutional Review Board (DF/HCC Protocol 17–102) on 05/30/2017. For access to full study protocol, consent documents, IRB approvals, and SPIRIT checklist see Supporting information section.

**Participant timeline.**   Families (patient and one of the parents) are enrolled for a total of 18 weeks (2 weeks run-in and 16 weeks follow-up). Fig 1 shows the enrollment, interventions, and assessments schedule.

**Interventions.**   After an initial 2-week run-in period, families who answer ≥2 PediQUEST surveys per participant (responders), are randomized (1:1) to PediQUEST Response or usual cancer care arms (week 0).

*PediQUEST (PQ) Response Intervention*. PQ Response is a multicomponent intervention designed to activate the symptom management process through two core components:

**Table 1.  PediQUEST response eligibility criteria.**

| Inclusion Criteria (Eligible patients must comply with ALL the following criteria) | Exclusion Criteria (Parent-patient dyads will be excluded if ANY of the following apply) |
|---|---|
| • ≥ 2 years old, <br>• receiving ongoing care at one of the participating sites, AND not "in remission <u>and</u> off cancer-directed treatment," <br>• have advanced cancer, defined as: at least a 2-week history of progressive, recurrent, or non-responsive cancer of any type, or any brainstem tumor, or a grade IV Glioblastoma Multiforme, or decision not to pursue further cancer-directed therapy, OR any other progressive/recurrent brain or solid tumor, <br>• be palliative care (PC) naïve, defined as the PC team not currently integrated into their regular care (operationalized as: ≤2 prior contacts with the PC team if these occurred ≥2 months ago and no plans for continued follow-up in place, OR if >2 prior encounters with the PC team last contact should have been ≥ 6 months ago and no follow-up plans in place). | • patient, is older than 18 years of age and none of their parents are involved in their care, OR has a non-brainstem low-grade glioma with localized progression/ relapse only, OR is expected to receive a stem cell transplant within 18 weeks of enrollment, OR is not expected to survive at least 2 months after enrollment; OR <br>• both parents, are foster parents who do not have legal guardianship, OR do not speak English or Spanish, OR are unable to understand and complete surveys. |

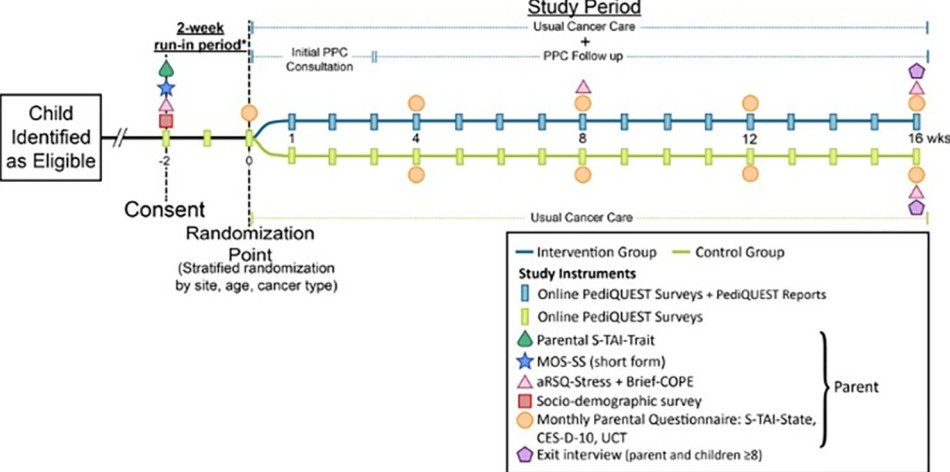

**Fig 1. PediQUEST response trial overview.** *After consent there is a two-week run-in period to identify (and exclude) non-responders (i.e., dyads who answer <2 PQ-Surveys/participant). *Abbreviations*: S-TAI: Spielberger's State-Trait Anxiety Inventory (two components: State and trait); MOS-SS: Medical Outcomes Study-Social Support; CES-D-10: Center for epidemiologic Studies Short Depression Scale; UCT: Use of Complementary Therapies Checklist; aRSQ-Stress: Adapted Response to Stress Questionnaire (stress portion).

**PQ web system (Surveys + Reports + email).** The PQ web system can be accessed through any electronic device via web or mobile application (App). Families assigned to the intervention have access to the full PQ system which consists of: (a) *PQ-Surveys*: measuring symptom burden and HRQoL. Surveys are automatically assigned on a weekly basis to all parents and children (if ≥5 years old); automated reminders are sent daily for two days; after 48 hours, unanswered or incomplete surveys are auto-submitted; (b) *PQ-feedback report*: generated automatically when PQ Surveys are answered. Provides a graphic summary by respondent (child and/or parent) of up to three-months of patient HRQOL and symptom scores and distress level for each symptom. A summary section compares current findings to prior survey; (c) *PQ emails/notifications*: a pdf of the report is automatically emailed—and made available through the App—to designated recipients, including the child if > 8 years old, parents, primary oncology team (including doctor, nurse practitioner and psychosocial clinician), and a designated PC team member, usually a nurse practitioner. PQ reports are the vehicle for a patient-mediated activation of the symptom management process.

**Early Integration of the PC (Response) team**: Integrated PC is provided through clinicians who typically provide PC consultation in the institution and serve as the "Response team." All participating Response teams receive an in-person 2-half day intervention-specific training before the study opened at each site. Recommended activities for Response team involvement are described in Fig 2 and include an initial in person or virtual (added after March 2020) consultation with the family within 3 weeks of randomization followed by phone, videoconference or in person contacts with the family: (i) on a regular basis over the intervention period (we encourage monthly encounters), (ii) to respond to PQ reports if distress is noted, and (iii) to monitor treatment recommendations. The goal is to *provide interdisciplinary consultative care focused on symptom and HRQoL assessment, prevention, and treatment* that can be delivered directly or through primary team clinicians. Ultimately, the primary oncology team (P.O.T.) *holds the decision to implement recommendations*. All communications and visits are documented in the patient medical record. Response teams are expected to maintain ongoing and direct communication with the P.O.T. throughout the follow-up period. Once the

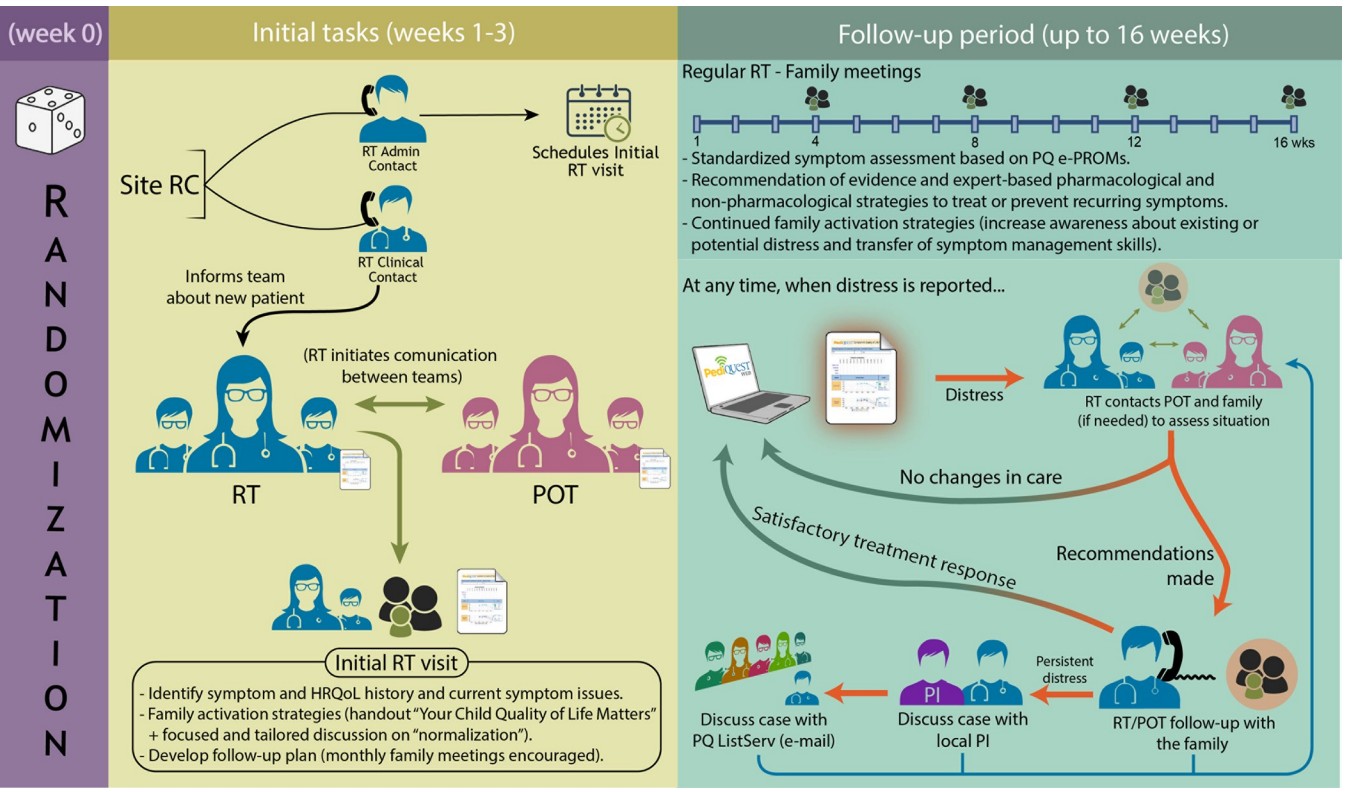

**Fig 2. Main activities of the response team integration component.** *Abbreviations*: *RC*: Research coordinator; *RT*: Response Team; *POT*: Primary Oncology Team; *PQ e-PROMS*: PediQUEST electronic patient reported outcome measures; *Local PI*: Site Principal Investigator; *PQ ListServ*: PediQUEST Listserv (expert listserv).

proposed follow-up is completed, participants in the intervention arm are offered the opportunity to continue being followed by the PC team. In addition, a study listserv including an interdisciplinary roster of national PC experts is available to all Response team members to ask for symptom management advice. PQ Response listserv member guidelines follow the American Academy of Pediatrics—Pediatric Hospice and Palliative Medicine (PHPM) LISTSERV® posting guidelines and the American Academy of Hospice and Palliative Medicine "Connect" Access Agreement.

*Usual cancer care*. Participants assigned to the usual cancer care arm receive the usual cancer care provided by participating sites and are required to complete PediQUEST surveys, but no reports are generated. These patients may receive regular PC consultations following the site's usual referral procedures.

*Rationale for choice of comparator group*: The goal of this study is to evaluate whether PediQUEST Response can improve current practice. For this reason, we chose a "usual cancer care" control group, which we hope represents the care that patients and families receive on a day-to-day basis. We are aware that this choice involves some potential for bias, as patients and providers may become influenced as the trial develops. However, we believe the risk for bias is low and we are taking some actions to minimize this risk (see Discussion).

*Adherence*. Intervention fidelity is monitored following NIH Behavior Change Consortium's recommendations [45]. Indicators include successful provider training, Response teams' adherence to study procedures, and participants' adherence (Table 2). *Response teams' adherence* to intervention delivery is promoted during the initial training session and

**Table 2. Intervention fidelity indicators.**

| Process | | Indicators | Source | Period | Target |
|---|---|---|---|---|---|
| RT Intervention components | Initial RT Meeting | 1. No. times visit occurs within 3 weeks randomization/No. randomized to intervention | MR | Monthly | >90% |
| | | 2. No. times symptom history addressed* in initial or next visit / No. randomized to intervention | | | >90% |
| | | 3. No. times "Your Child QOL Matters" material handed to family and discussed*/No. randomized to intervention | | | >80% |
| | | 4. No. times RT communication with primary team documented*/ No. randomized to intervention | | | >90% |
| | Follow-up | 1. No. contacts of RT w/family or POT triggered by distress/No. Surveys w/ distress | MR + PQ Reports | Monthly | >80% |
| | | 2. No. RT notes containing at least 1 recommendation (treatment or preventive, pharmacologic or non-pharmacologic), or otherwise providing the basis for no change in treatment/No. Surveys w/ distress | | | >90% |
| | | 3. No. times RT communication with POT documented /No. contacts with family | | | >75% |
| RT Training and clinical support | | 1. No. providers with complete training (face to face + evaluation) / No. of PC providers /site | Training logs | Once | >80% |
| | | 2. No. consults to ListServ / No. episodes of persistent distress/month | PQ Reports + ListServ logs | Monthly | >80% |

Abbreviations: No.: Number; QOL: Quality of Life; RT: Response team; POT: Primary Oncology Team; MR: Medical Record.

*The Study Data Center provided a set of smart text phrases to facilitate the documentation of intervention processes. All PC notes from intervention patients will be abstracted and saved as pdf. Indicators will be built from the PC notes.

thereafter, during their regular meetings with local PIs. Participant PC teams receive a monetary incentive per subject assigned to the intervention. To monitor Response team's adherence, we regularly collect PC notes on intervention arm participants and triangulate them with distress episodes reported through PediQUEST, intervention delivery metrics (e.g., number of PC encounters), and use of listserv. This compiled information is shared with the sites one time during recruitment after at least 10 patients receive the intervention (or one year has elapsed since starting recruitment) to monitor and improve intervention delivery. *Participants' adherence* to study procedures is facilitated by a relatively short follow-up, PQ system automated reminders, and study team close monitoring of survey response. Participants who miss surveys are contacted by the RCs to identify difficulties and willingness to continue participation. If either parent or child misses four consecutive PediQUEST surveys, the dyad will be removed from the study (eliminated). Rare exceptions to this criterion may be granted at the discretion of the PI and PM. *Qualitative fidelity evaluation*: Patients and parents, and a convenient sample of providers are invited to participate in *exit interviews* to talk about their experience with the study and provide more details about intervention delivery.

**Outcomes.** *Primary Outcome Measures. Child Quality of Life*: Difference between the two treatment arms in the mean over 16-weeks of Pediatric Quality of Life Inventory 4.0 (PedsQL) total score as reported by (a) the parent for all enrolled patients, and (b) the patient if 5 years of age or older.

*Secondary Outcome Measures*

*Patient-level outcomes*

*Child Quality of Life Domains*: Difference between the two treatment arms in the mean over 16-weeks of PedsQL sub-scale scores (physical and psychosocial) as reported by (a) the parent for all enrolled patients, and (b) the patient if 5 years of age or older.

*Symptom burden scores*: Difference between the two treatment arms in the mean over 16-weeks of PQ-MSAS total and sub-scale scores (PHYS and PSYCH sub-scales) as reported by (a) the parent of all enrolled patients and (b) the patient when 13 years old or older. Child-

reported data using PQ-MSAS 7–12 will be used to conduct exploratory analysis (see rationale for the age limit in the Discussion).

*Parent-level outcomes*

*Parent Psychological Distress*: Difference between the two treatment arms in the anxiety and depressive symptom scores measured monthly over 16-weeks by the State-Trait Anxiety Inventory- State component (S-TAI-State) and the Center for Epidemiological Studies Short Form (CES-D-10); and on symptom-related stress measured at study entry and at week 16 by the adapted version of the stressor portion of the Response to Stress Questionnaire (*a*RSQ-Stress).

*Family activation*: (a) Change in active coping, planning, and instrumental support Brief-COPE scale scores between week 16 and baseline, ordinal variables; (b) Total number and (c) number of different complementary therapies used for symptom treatment (reported by parents every 4 weeks), and (d) use of psychosocial services, as measured by total number of psychosocial clinician encounters documented in the medical record, over the 16-weeks study period.

*Sample Size and Power Considerations*

Based on prior work [35], we estimate that there will be approximately 32 eligible patients/site/year, representing a total of 288 eligible patients available for recruitment over the 36 months of the study. Pilot data suggest an enrollment rate of 70% which translates into approximately *200 patient-parent dyads*, our *enrollment target*. If 25% are non-responders, a conservative estimate based on pilot data, 150 patients will be randomized (n = 75/arm). We expect an attrition of approximately 10%, mainly due to dropouts, which will lead to complete data on *a minimum of 136 dyads*, *68 per group*. Although the sample size is mainly driven by pragmatic considerations (target population is small), power calculations were conducted and indicate that our sample size of 68 children per group provide sufficient power ($\geq$80%) for both primary and secondary outcomes for clinically meaningful effect sizes. For patient-level and parent distress outcomes power estimates are conservative because they only use one observation per subject (average across time) without accounting for the repeated measures within subject. For parent-level outcomes (distress and activation) we report the power to detect moderate to large effect sizes. Power calculations are based on two sample independent one-side tests, $\alpha$ = 0.05 unless otherwise specified.

*Power calculation for patient-level outcomes (HRQoL (primary) and Symptom (secondary))*: Standard deviations were estimated using data from our PQ study (39 weeks follow up, surveys answered when attending clinic or once a month) [34]. For each child, we identified all 16-weeks periods with $\geq$6 surveys and calculated PedsQL and PQ-MSAS mean total scores for each 16-weeks period. We randomly selected one period per child and estimated the standard deviation (SD) of 16-weeks PedsQL and PQ-MSAS means. This strategy was repeated 10 times and a pooled estimate was calculated. A sample of 136 children (68 per group) achieves 85% power to detect a 4.4-point increase minimal clinically important difference (MCID) in the mean over 16-weeks PedsQL total score (SD = 9.5, for both groups), and a 95% power to detect a 3-point decrease in mean over 16-weeks PQ-MSAS total score (SD = 5.3). Power calculations are conservative as the SD estimates are based on mixed responses from parents and children, while in the current proposal responses from different informants will be considered in separate analyses and are expected to have less variability.

*Power calculation for parent-level outcomes–Parent distress (Parental anxiety (S-TAI-State), and depressive symptoms (CES-D-10), secondary outcomes)*: A sample of 68 parents per group achieves > 80% power to detect a moderate effect size (Cohen's d = 0.43). For example, this effect size corresponds to a difference of 5.2 points in mean S-TAI-State (SD = 12.2) [46].

*Power calculation for parent level outcomes—Family Activation* (*secondary outcomes*): Change in outcome level between baseline and 16-weeks: the target sample size achieves 80% power to detect an effect size of 0.5 in any of the coping scale scores. This effect size corresponds, for example, to a score change of 0.75 points between baseline and 16-weeks (SD = 1.5, consistent with literature reports) [47]. The sample size has 82% power to detect a mean increase of 2 encounters with psychosocial clinicians between trial arms across 16-weeks (SD = 4.5, from prior PQ Study). Finally, for mean number of complementary therapies used for symptom treatment over 16-weeks, our sample size has 80% power to detect an effect size of 0.43.

*Recruitment*

Research coordinators (RCs) at each center regularly review clinic rosters, e-mail sign outs, attend clinic or tumor board conferences. All patients ≥ 2 years old AND seen by the outpatient oncology clinic or inpatient oncology service (base population) are entered into a *pool list excel file* that has automated fields allowing for a quick pre-screening process. For all patients pre-screened as potentially eligible, RCs further review the medical chart to determine eligibility and have eligibility validated by the local PIs (or designated investigator). If a patient is eligible, the RC sends an email to the primary oncology provider informing them that the patient will be invited to enroll unless the provider opts the patient out. The entire process is registered in a REDCap-based screening and tracking system [48] that includes automated steps such as emails and reminders to prevent the inadvertent loss of potential participants. Initially, we approached families in person at the clinic or ward. In March 2020, because of the COVID-19 pandemic, enrollment was halted for three months in compliance with sites regulations. Enrollment resumed between June and October 2020 at the sites. Due to COVID restrictions, we added a virtual enrollment option consisting of an initial email followed by a videoconference or phone meeting to hold the enrollment conversation with interested families.

Strategies to enhance recruitment include in-service sessions with providers to reduce potential gatekeeping, a direct family approach, flexibility in scheduling recruitment interviews, and working with experienced RCs. In addition, it is expected that the study's relatively short duration, RCs facilitation of PediQUEST web's registration and close monitoring throughout the study, and the small non-monetary incentives ($40 USD monthly gift cards for the patient and participating parent), boost recruitment and retention.

## Assignment of interventions (allocation and blinding)

Participants are randomly assigned to the intervention or control group with a 1:1 allocation as per a computer-generated randomization sequence created by the study statistician, and stratified by site, age (2–7, 8–12, ≥13 years old), and type of cancer (hematological, non-hematological malignancies). Random sequences are embedded in REDCap by informatics personnel and concealed to research staff. After the run-in period, if the dyad is classified as being a responder, study arm assignment becomes visible to RC in REDCap. After randomization, participants, PC team, oncologists, and research staff are unblinded to group assignment, because the intervention is not amenable for blinding.

## Data collection, management, and analysis

**Data collection methods.** Participant reported data are collected through the PediQUEST web system. Research personnel help families register in the system and download the mobile App. All participating parents and patients ≥8 years old are assigned a PQ account, linked to an email of choice, to receive the surveys and see the reports (if in intervention group). Children aged 2 to 4 years old do not answer any questionnaires and children 5 to 7 years old only

answer the HRQoL questionnaire; 7-year-olds also respond to the PQ-MSAS which is read out loud by their parents. All parents answer proxy versions of the questionnaires. Participants without home internet (estimated at <10%) are provided a tablet with cellular service during their participation in the study.

*Instruments*. Table 3 summarizes the main characteristics of study instruments. After registration, research personnel manually assign the *Baseline Packet*. Over a total of 18 weeks (2-week run-in period and 16-weeks post-randomization follow-up), parents and patients ≥5 years old in both arms are automatically assigned the corresponding version of the weekly *PediQUEST survey*. In addition, parents of both arms are assigned *Monthly Parent Questionnaires* evaluating distress, burden, and use of complementary therapies, beginning at the time of randomization (Fig 1). *Parent Activation Surveys* are administered at weeks 8 and 16 (the measure at week 8 will be used for an exploratory mediation analysis).

*Other data sources*. *Medical records* are abstracted by research personnel to gather information about child diagnosis and clinical status at study entry, and disease status (stable, progression, remission), date of change in status when appropriate, cancer-directed treatments and dates, formal PC referrals, hospitalizations, number of psychosocial clinician encounters and number of PC encounters documented throughout the follow-up period.

All patients ≥8 years old and parents who complete the 16-weeks follow-up or dropped out, parents of children who died, and a convenient sample of oncology and PC providers, are invited to participate in an *Exit Interview* (face-to-face or phone interview) administered by RCs to evaluate study processes, care experience, and intervention fidelity (if in the intervention arm).

**Data management.** Each eligible participant is assigned a unique study identification number. Most study data are collected and stored electronically through REDCap [48] and the PQ web system. All other source documents, apart from consent documents, are created as or converted to electronic format and saved in HIPAA compliant server locations. These data are protected by robust security features including secure user authentication, password encryption in both front and back ends, and role-based access controls that prevent users from accessing data that they are not authorized to see. All study personnel collecting Personal Health Information have HIPAA Certification and the training mandated by the Institutional Review Board. Consent documents are stored in locked cabinets and accessible only to study team members. Data management is compliant with Good Clinical Practices [54].

**Analysis plan.** All statistical analysis will be conducted on an intention-to-treat basis. Baseline characteristics will be described and compared between groups using summary measures (e.g., means, medians, rates) and tests selected based on variable distribution. Analyses will be adjusted for factors used in the stratified randomization (site, age group and type of cancer). In addition, if, despite efforts to standardize intervention delivery, we identify substantial differences in fidelity across sites, we will evaluate effect modification by site including the site × intervention interaction in the models. The False Discovery Rate criterion will be used to correct for multiple testing in non-planned analyses. *Considerations regarding missing data*: For outcomes in which missing completely at random is a plausible assumption, we will use complete case analysis. If missing at random is highly likely, we will use multiple imputation or inverse probability weighting, depending on the statistical model being considered. If not, we will use sensitivity analyses. Robustness of the estimates to missing data assumptions will be assessed.

*Comparison of patient-level outcomes*: *HRQoL and Symptom distress*. The effect of the intervention on child quality of life and symptom burden outcomes will be estimated using mixed linear models, with patient as a random effect and group, time (categorical) and group x time interaction and the stratification factors as fixed effects. Treatment success will be defined as a

**Table 3. Study instruments.**

Most baseline tools will help characterize the study sample.

| Respondent | Construct | Instrument | Description |
|---|---|---|---|
| **Baseline Packet (week -2, right after consent)** | | | |
| **Enrolled Parent** | **Demographics** | Demographic section of the Survey about Caring for Children with Cancer (SCCC), originally developed by study investigators [34] | Age, gender, race, marital status, number of siblings (of participant child), religion, religiousness, education, zip code. |
| | **Household Material Hardship** | Household Material Hardship Survey [49] | Self-report family income, child's health insurance, health literacy, evaluation of household hardship (housing and transportation, utilities, food insecurity, and financial strain). |
| | **Social Support** | Medical Outcomes Study Social Support-Short Form [50] | 4 domains: emotional/ informational, tangible, affectionate, and positive social interaction; 8-items total. Response options: 5-point Likert type (from none of the time to most of the time). Scoring: sum of item scores. Score range: 8–40 (higher = more support). High reliability ($\alpha > 0.91$) and stability over time. |
| | **Anxiety** | Spielberger's State-Trait Anxiety Scale (S-TAI)-Trait subset [41] | Measures trait anxiety (20 items); evaluates relatively stable aspects such as calmness, confidence, and security. Response options: 4-point Likert type (from almost never to almost always; scoring reversed for some items). Scoring: sum of item scores. Score range: 20–80 (higher = greater anxiety). High validity, reliability, and stability over time. |
| | **Symptom-related stress** | Stress portion of the Response to Stress Questionnaire-Pain (aRSQ-stress) [43,51] | Adapted version including 9 items that evaluate parental stress in past month related to uncertainty, meaning, treatment adherence, and effects on daily life of child symptom distress; Response options: 5-point Likert type (from no stress at all to a lot of stress). Scoring: sum of item scores. Score range: 9–45 (higher = higher stress); 1 open-ended item that asks about other sources of stress; and 1 item that assesses overall perceived control over these problems; Response options: 5-point Likert type (from no control at all to a lot of control). Score range: 1–5 (higher = greater control). High internal consistency [52]. |
| | **Activation** | Brief-COPE [44] | Adapted version (15 items) of the Brief-COPE scale evaluating parent's use of five coping strategies relevant to symptom management using two items each: active coping, planning, instrumental support, acceptance, and self-blame. Five additional items evaluate emotional support, religion, positive reframing, behavioral disengagement, and denial. Scores on each scale (or item) range from 2 to 8, with higher scores indicating greater use of that strategy. Use of selected subscales encouraged by author. Use of single items decided after piloting instrument and finding no variability for the 2-item scales. |

*(Continued)*

**Table 3.** (Continued)

**PediQUEST Survey (weekly, from week -2 to week 16)**

English and Spanish **PQ-Surveys** which have 5 versions (respondent- and age-adapted) that collect **both** parent and child reports (child from the age of five).[1] Children answer the same survey version throughout the 16-weeks according to the age at time of enrollment.

| Instruments | PQ-Memorial Symptom Assessment Scale (PQ-MSAS) [19,39,40,53] | | | Pediatric Quality of Life Inventory Generic Core Module (PedsQL 4.0™) [37,38] | | | | | | | | Overall Well-being [34] | | |
|---|---|---|---|---|---|---|---|---|---|---|---|---|---|---|
| Construct measured | Symptom Burden | | | HRQOL | | | | | | | | General Health Perception | | |
| Instrument Characteristics | Measures presence, severity, frequency, and extent of bother in past weeks for 26 physical, psychological and 3 "other" symptoms | | | Assesses how much of a problem each item has been in past week (evaluates physical (7 items), and emotional, social, and school performance (5 items each)) | | | | | | | | 1-item ("Overall, how have you been feeling during the past week?" Anchors: not well at all; extremely well) | | |
| Versions | Proxy (Parent) | Self-report | | Proxy (Parent) | | | | Self-Report | | | | Proxy (Parent) | Self-report | |
| | | 7-12[2] | 13-18 | 2-4 | 5-7 | 8-12 | 13+ | FS-3 | 5-7 | 8-12 | 13+ | | 5-6 | >6 |
| Response Types[3] | L-5 | L-4 | L-5 | L-5 | | | | FS-3 | L-5 | | | VAS | FS-3 | VAS |
| PQ Versions | 2-4 | | | | | | | | | | | | | |
| | 5-6 | | | | | | | | | | | | | |
| | 7 | | | | | | | | | | | | | |
| | 8-12 | | | | | | | | | | | | | |
| | 13+ | | | | | | | | | | | | | |
| Validation data | high reliability (α = 0.81)[4] | | | high reliability (α = 0.93 parent and 0.88 child report) | | | | | | | | Spearman Correlation with PQ-MSAS = -0.56[§] | | |
| Scoring range | 0-100 (100 = worse) | | | 0-100 (100 = better) | | | | | | | | 0-100 (100 = better) | | |

**Monthly Parent Questionnaires (weeks 0-4-8-12-16)**

| Respondent | Construct | Instrument | Description |
|---|---|---|---|
| Enrolled Parent | Anxiety | Spielberger's S-TAI-State subset [41] | State-anxiety (20 items) (sensitive to transient changes) measures presence and severity of current symptoms of anxiety. Response options and scoring same as for STAI-T. Suggested cut-point: 39-40. High validity and reliability (α = 0.86-0.95) |
| | Depressive symptoms | Center for Epidemiologic Studies–Short Depression Scale (CES-D-10) [42] | Assesses frequency of occurrence during the past week of depressive symptoms (10 items). Response options: 4-point Likert type (from rarely or none of the time to most or all of the time; scoring reversed for two items). Scoring: sum of item scores. Score range: 0-30. (higher = depressed mood). Suggested cut-point: 10. High reliability (α = 0.88-0.91) |
| | Symptom-related stress | aRSQ-Stress [43,51] | See above. Measured at weeks 8 (for exploratory mediation analysis) and 16 |
| | Activation | Brief-COPE [44] | See above. Measured at weeks 8 (for exploratory mediation analysis) and 16 |
| | | Use of complementary therapies checklist | Ad hoc checklist of complementary therapies; parents report use (yes/no) in past four weeks and reasons for use. |

[1] Data from 5–12 year-olds will mainly be used for intervention delivery purposes (PediQUEST reports provide feedback on child and parent answers when available) and to analyze patient-parent concordance (see Discussion)

[2] PQ-MSAS 7–12 evaluates shorter time frame (past two days) and only 8 symptoms

[3] Response types: L-5: 5-option Likert type; L-4: 4-option Likert type; FS-3: 3-option Faces scale; VAS: Visual analogue scale (100 mm)

[4] Data from pilot PQ RCT [34].

mean difference between groups of at least 4.4 points, the MCID for PedsQL [38]. The same analysis will be undertaken for the respective domain subscales of PedsQL (physical and psychosocial; MCID is 6.7 and 5.3 points for each subscale respectively [38]) and PQ-MSAS (physical and psychological).

*Comparison of parent-level outcomes*: *anxiety, depressive symptoms, and burden.* The same analytical approach described for patient-level outcomes will be followed to analyze parent S-TAI-State, CES-D-10 mean scores and aRSQ symptom-related scores. In S-TAI-State analysis, we will additionally adjust for baseline S-TAI-Trait score. We will also compare depressive symptoms (CES-D-10 $\geq$10) over time between the two groups using generalized linear mixed models with logit link and binomial distribution. As (parent) gender is a predictor of these outcomes [55], it will be included in these models if imbalanced between arms.

*Comparison of parent-level outcomes*: *family activation.* Mixed models, like those described for patient-level outcomes, will be fitted to estimate the effect of the intervention on change between week 16 and baseline in Brief COPE active coping, planning, and instrumental support Scores, Total number of, and number of different, complementary therapies, and number of psychosocial clinician encounters over the 16-weeks study period. Link and distribution for these models will be defined based on the distribution of the outcome variable. We will explore whether change in parent's activation score is modified by parent/child characteristics (e.g., parent's gender, S-TAI-Trait, CES-D-10 scores; and child's diagnosis, among others) including the corresponding interaction terms.

*Additional Analyses.* We will also conduct a thorough process evaluation triangulating (a) intervention fidelity indicators (Table 2); and (b) exit interviews to better understand whether and how processes were influenced by (or influenced) the intervention. Quantitative data will be reported using descriptive techniques. Qualitative data will be transcribed verbatim, personal identifiers removed, and coded using an inductive, open, and iterative coding strategy to fit the data. Once consensus is reached, a coding dictionary will be developed, and the data recoded independently by two researchers. There will be two analysis levels: 1. individual and 2. cross-case analysis (i.e., grouped by age and site). Transcripts will be analyzed with a thematic analysis [56] approach using MaxQDA software [57].

Finally, and depending on the results of the main analysis, we will also conduct a series of exploratory analyses including assessing parent activation as a mediator in the intervention effect on patient's HRQoL, evaluating age as a potential effect modifier, and, if sample size permits, a subgroup analysis of PQ-Response effects on symptom burden as reported by children aged 7–12 years.

## Data monitoring

**Monitoring.**  This is a small multisite clinical trial of a supportive care intervention with a risk profile that is comparable to usual cancer care. As such, data monitoring is primarily carried out by the Study Data Center (SDC) at Dana-Farber Cancer Institute, and a small Data and Safety Monitoring Board (DSMB) consisting of 5-members representing different disciplines and expertise (three members are unrelated to the study and act as Independent Monitors). The DSMB holds quarterly conference calls to provide input and guidance on participant's accrual, retention and safety, data quality, compliance issues, intervention fidelity, and protocol deviations or violations. A blinded primary end-point interim analysis was conducted by the study statistician at 12 and 24 months after the study opened primarily to analyze data variability. The DSMB will present recommendations to the Steering Committee as appropriate.

**Potential risks and benefits.**  *Burden from answering study surveys* is expected to be minimal. Should a patient or parent become upset after completing any of the study assessments or

during the intervention, they may stop answering/participating and a consultation with a clinician from the corresponding Psychosocial Oncology Program is offered.

*Risk resulting from randomization*: For example, not being exposed to the Response intervention (control arm) may result in patients having persistent distress, or, on the other hand, being exposed to the Response intervention may result in side effects related to medications or complementary therapies recommended for symptom management; however, none of the expected risks are above and beyond what a patient receiving cancer care may experience.

*Potential for Confidentiality breach*: Because of the safeguards in place, which include among others, a Certificate of Confidentiality from the NIH, we believe that the risk of serious breaches is extremely low. See Confidentiality section.

*Safety monitoring* is the responsibility of the SDC and the DSMB. Information on all potential types of adverse events is collected on an ongoing basis and recorded on standard forms. If at any point a serious adverse event is detected (such as a death, suicide or serious consideration of suicide that is felt to be related to a recommendation of the Response team), we will suspend study activities and try to determine if there is any link between study procedures and the adverse event and determine if any modification is advised or if the study should be stopped.

**Auditing.**   Electronic data captured through REDCap and PediQUEST web is subject to electronic validation and is cross validated by our staff for complex errors and completeness twice a week. Regular site monitoring includes biweekly calls between the Project Manager (PM) and sites' RCs to oversee recruitment and protocol adherence (recurrence of these meetings is flexible adjusted to study needs). Study progress reports are reviewed by the PM, the DSMB, and by the Steering Committee during their quarterly meetings. Whenever data quality or trial progress problems, or protocol violations are detected, the local PI is contacted. If the problem is severe or persistent, a site visit will be scheduled. The funding agency and intervening IRBs will be notified in compliance with each IRB's regulations.

## Ethics and dissemination

**Research ethics approval.**   Protocol, template informed consent forms, participant's materials, and any other requested documents as well as all subsequent modifications, were reviewed and approved by the sponsor and the applicable Institutional Review Boards with respect to scientific content and compliance with applicable research and human subjects' regulations. Site IRBs provided a waiver of individual authorization for disclosure of personal health information for the identification process and a waiver of documentation of consent for adolescents who reach the age of majority while on study, PC, and oncology providers. The current protocol was initially approved by the DF/HCC IRB (Protocol 17–102) on 05/30/2017 and has since been continually reviewed and approved by the same IRB (current version is Amendment 18 / Version 17.1, approved on 04/21/2022).

**Consent/Assent.**   Patient assent and parental written permission is secured by the RC before formally entering the study. Study brochures and informed permission/assent document contain all the required elements of informed consent including the purpose of the study and procedures, and potential risks and benefits of participation. The forms are available in English and Spanish. Interpreters are used to aid with the consent process for Spanish speaking families if needed. At least one parent is asked to review and sign the informed permission/assent documents covering the patient's participation. For children between 5 and 9 years of age the study brochure is considered the assent document and verbal assent is sought whenever possible and if developmentally appropriate. Children and adolescents between 10 and 17 years of age are invited to review and sign a developmentally adapted assent document (there

are two assent documents, one for 10–12 years old and one for teenagers). Patients ≥18 years old are invited to review and provide written consent covering their participation. A separate consent document covers caregiver's participation. Because of the COVID-19 pandemic, four IRBs allowed verbal in place of written documentation of consent/assent.

**Confidentiality.** To carry out the study it is necessary to collect and store some personal health information (PHI) including contact information, patient's date of birth and death, if applicable, and parent's date of birth. These data are collected during the recruitment process through REDCap and PediQUEST web. Beyond the description in the Data Management section, these Good Clinical Practices and HIPAA compliant systems use digital certificates to validate the identity of trusted partners and keep a full audit trail of all transactions. Access outside the SDC firewall occurs through a send secure system. All data is stored on secure servers specifically allocated to the study. Audio recordings and transcripts are stored as encrypted files on a password protected web-based repository. Access to all data is limited to study personnel on a "need to know" basis. Data analysts have access to a limited dataset that contains the minimal amount of PHI that is essential to conduct the analyses.

**Dissemination policy.** Analysis and reports of the PediQUEST Response trial results will include the full sample, unless scientifically justified. The Steering Committee will make recommendations regarding when and what material should be submitted for publication. Each paper will be reviewed and approved by the SC members prior to submission. The SC will work to reduce the interval between end of data collection and release of the study results. We expect to take about 4 to 6 months to compile and submit the main paper. Study results will be released to the participating physicians and referring physicians through publications in peer-reviewed journals, conference abstracts, and oral presentations, whereas specific materials will be produced to disseminate results among study participants and the general public. Results will also be available in clinicaltrials.gov.

# Discussion

This study will examine whether PQ-Response, an intervention that combines the collection and feedback of e-PROMS with integration of PC in the care of children/AYAs with advanced cancer, improves child and parent distress. It will be one of the first RCTs to study the effectiveness of early integration of PC in children/AYAs.

## Study rationale

The study is designed with a strong pragmatic [58] emphasis: we use existing PC clinicians to deliver the intervention, do not require extra visits, and have chosen outcomes that are highly relevant to participants. We expect this design will help understand how to integrate pediatric PC services in real life settings.

*Intervention's "dose," frequency, and administration*: *Weekly PQ-Surveys*: Survey frequency was decided based on pragmatic reasons. Clinically, the weekly timeframe is practical to both to capture distress in real time (longer periods may affect recall) and to allow the Response teams to intervene and propose preventive strategies if symptoms are recurrent. Adherence to weekly surveys was good during the pilot study (manuscript in preparation). We considered allowing for survey completion on an "as needed" basis, to better capture acute distress, however, this approach was considered challenging for controls who may have less incentive to report. *Response team intervention*: Regular contacts are proposed as a means of standardizing the intervention across patients with varying experiences of distress. During training we also provided keys to interpreting PQ reports and scores to promote optimal Response team

intervention. Further, this approach helps standardize the response to distress and increases replicability of the intervention.

*Why use parent and patient reports to measure patient outcomes*? While self-report remains the gold-standard for patient HRQoL and symptom burden outcomes, there is an increasing body of evidence suggesting that there is value in having both parent and child/AYAs reports [59] as each provide a unique perspective [60]. Together, these reports provide a broader picture of the child/AYAs' situation [61]. Yet, one of the consequences of using "multiple informants" [62] is that the information may be non-concordant (divergent) [63,64]. The optimal approach to consolidate multiple informant responses into a unique child/AYAs outcome is still unresolved [60]. Acknowledging the clinical importance of considering both "voices" (i.e. patient and parent), we measure patient outcomes (HRQoL and symptom burden) using both reporters. The chosen instruments allow parental report across the study's full age range, and patient self-report from the age of five for HRQoL, and 7 for symptom burden. Of note, because the younger child version of PQ-MSAS for ages 7–12 does not measure the full range of symptoms we will not use it as an outcome measure, rather it is used as part of the intervention and for exploratory purposes. Given the lack of consensus as to how to handle multiple informant information, the effects of the intervention on patient outcomes will be analyzed separately for each of these informants.

## Anticipated problems and solutions

Anticipated potential problems may include *failure to meet sample target*, *high attrition*, *contamination* in usual care arm, *feasibility*, and *social desirability bias*. Given the strategies presented earlier, we expect strong recruitment rates. From prior experience, attrition due to death and drop-out are expected to be low and were accounted for in sample size calculations. However, we are collecting socio-demographic data and reasons for dropout or elimination to better understand recruitment and attrition drivers. As we successfully did in prior studies [35], PI and Co-Is closely monitor recruitment and retention rates and characteristics. When concern arises, we revise strategies together and problem solve. Contamination risk is expected to be very low. Even so, we will evaluate the proportion of clinicians who provided care to trial participants in both arms and evaluate behavior shifts during exit interviews, to understand whether contamination was an issue. We believe that the PQ Response intervention is highly feasible: (a) It builds on our prior successful research work; (b) all participating sites have long-standing working relationships and very similar oncology care and PC team approaches; and, (c) the intervention has been carefully piloted and suggests feasibility. Social desirability bias is a common issue in un-blinded studies. We plan to minimize this risk by not emphasizing the specific outcomes we are targeting, and by careful training and supervision of RCs.

## Trial status

The PQ-Response trial started recruiting on 04/01/2018 and anticipates finalizing data collection on July 1, 2022.

## Supporting information

**S1 Checklist. SPIRIT checklist.**
(PDF)

**S1 Protocol. PediQUEST Response Study Protocol V 17.1.**
(PDF)

**S2 Protocol. Consent documents.**
(PDF)

**S3 Protocol. IRB Approval of Study Protocol (original).**
(PDF)

**S4 Protocol. IRB Approval of Study Protocol (current).**
(PDF)

## Acknowledgments

The authors gratefully acknowledge children/AYAs, parents, and health care providers for their willingness to contribute their time to the study. We want to extend a special thanks to our study collaborators (listed below) including site investigators, research staff, data management teams, and Data and Safety Monitoring Board members for their commitment to conduct and implement the study, recruiting efforts, dedication to the quantitative and qualitative data collection process, and sensible guidance. Additional thanks to Stefan Friedrichsdorf, MD, Medical Director, Center of Pediatric Pain, Palliative and Integrative Medicine, and head of EPEC-Pediatrics for his expert advice and participation in the Response teams training and to Justin Baker, MD Chief, Division of Quality of Life and Palliative Care for his encouragement and wisdom.

### Study collaborators

Site Investigators: Christina Ullrich (PI-Dana FCI), Chris Feudtner (PI-Children's Hospital of Philadelphia), Jason Freedman (Co-PI- Children's Hospital of Philadelphia), Abby Rosenberg (PI-Seattle Children's Hospital), Ross Hays (Co-PI- Seattle Children's Hospital), Elisha Waldman (PI- Ann and Robert H. Lurie Children's Hospital of Chicago), and Tammy Kang (PI–Texas Children's Hospital).

Research staff: Sarah Stevens, Leah Beight, Gabrielle Helton, Alexandra Merz, Nicole Etsekson, Kelly Shipman, Dane Olson, Liam Comiskey, Alison O'Daffer, Jen Chapman, Karen Crew, Mehdi Youbi, Connor Drakeley, Aneta Jedraszko, Tatiana Arevalo Soriano, Walia Namrata, Shimei Nelapati, Samantha Hurtado, Mellody Hellstein, Marissa Fury, Jessica Casas, Katie Amsdem, Lindsey Gurganious, and Erin Kim.

Data management teams: Rocio Rodriguez, Luz Gibbons, and Alvaro Ciganda from the REDCap Team, and Madhuri Deodhar, Datta Sukalkar, Preeti Joshi, and Deepali Karmalkar from the PediQUEST Team.

Data and Safety Monitoring Board members: Marie Bakitas, DNSc, NP-C, FAAN, Professor, School of Nursing/Department of Medicine, UAB The University of Alabama at Birmingham; Erin Currie, MSN, RN, PhD, CPLC, Adjunct Assistant Professor, School of Nursing/ Department of Medicine, UAB The University of Alabama; Mary Cooley, PhD, Rn, FAAN, Lecturer, Harvard Medical School; and Cynthia Gerhardt, PhD, Professor, The Ohio State University College of Medicine.

## Author Contributions

**Conceptualization:** Veronica Dussel, Liliana Orellana, Joanne Wolfe.

**Data curation:** Veronica Dussel, Rachel Holder, Madeline Avery.

**Funding acquisition:** Veronica Dussel.

**Methodology:** Veronica Dussel, Liliana Orellana, Joanne Wolfe.

**Project administration:** Madeline Avery.

**Resources:** Rachel Holder, Rachel Porth, Madeline Avery, Joanne Wolfe.

**Software:** Veronica Dussel.

**Supervision:** Madeline Avery.

**Visualization:** Veronica Dussel, Joanne Wolfe.

**Writing – original draft:** Veronica Dussel, Liliana Orellana, Rachel Holder, Rachel Porth.

**Writing – review & editing:** Veronica Dussel, Liliana Orellana, Rachel Holder, Rachel Porth, Madeline Avery, Joanne Wolfe.

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
