## [Decision Letter · Decision Letter 0]

24 Oct 2022

A multisite randomized controlled trial of an early palliative care intervention in children with advanced cancer: the PediQUEST Response Study Protocol

PONE-D-22-16245

Dear Dr. Dussel,

We’re pleased to inform you that your manuscript has been judged scientifically suitable for publication and will be formally accepted for publication once it meets all outstanding technical requirements.

Kind regards,

Doris YP Leung

Academic Editor

PLOS ONE

Additional Editor Comments:

The study reports the protocol of testing an early palliative care intervention in children with advanced cancer. The topic is very important and will have great clinical implications. I agree with the reviewer that there are some minor points for clarifications. Overall, the RCT is well-designed.

Reviewers' comments:

Reviewer's Responses to Questions

**Comments to the Author**

1. Does the manuscript provide a valid rationale for the proposed study, with clearly identified and justified research questions?

Reviewer #1: Yes

2. Is the protocol technically sound and planned in a manner that will lead to a meaningful outcome and allow testing the stated hypotheses?

Reviewer #1: Yes

3. Is the methodology feasible and described in sufficient detail to allow the work to be replicable?

Reviewer #1: Yes

4. Have the authors described where all data underlying the findings will be made available when the study is complete?

Reviewer #1: Yes

5. Is the manuscript presented in an intelligible fashion and written in standard English?

Reviewer #1: Yes

6. Review Comments to the Author

You may also provide optional suggestions and comments to authors that they might find helpful in planning their study.

Reviewer #1: This study protocol for the PediQUEST Response Study is exemplary in its description of the study protocol. There is a clear, comprehensive, and sophisticated data analytic plan. A good rationale is provided for how to handle multiple informant information. Likewise, a reasonable discussion of contamination risk and the plan to address it.

I have only a few minor suggestions to make.

Page 8 line 161 Please name the 5 study sites.

Page 20 line 450 Mention Certificate of Confidentiality from NIH as another protection against breaches to confidentiality.

Page 23, line 526, please update manuscript in preparation if applicable.

7. PLOS authors have the option to publish the peer review history of their article (what does this mean?). If published, this will include your full peer review and any attached files.

Reviewer #1: No
